# Biodiversity Targets, SDGs and Health: A New Turn after the Coronavirus Pandemic?

Claire Lajaunie [1,2,*] and Serge Morand [3,4,5]

1 INSERM—LPED Laboratoire Population Environnement Développement (IRD-AMU), CEDEX 03, 13331 Marseille, France
2 Strathclyde Centre for Environmental Law and Governance (SCELG), Law School, Strathclyde University, Glasgow G1 1XQ, UK
3 CNRS ISEM—CIRAD ASTRE, Montpellier University, 34090 Montpellier, France; serge.morand@umontpellier.fr
4 Faculty of Veterinary Technology, Kasetsart University, Bangkok 10220, Thailand
5 Faculty of Tropical Medicine, Mahidol University, Bangkok 10400, Thailand
* Correspondence: claire.lajaunie@inserm.fr

**Abstract:** In light of the coronavirus pandemic, we invite readers to a reflection over the aim and use of Sustainable Development Goals (SDGs), the determination of the new biodiversity targets in relation to health issues. Starting with a brief overview of the initiatives to consider health and the environment in the international arena before the adoption of SDGs, we show how the pandemic shed a new light on the need for research on the interlinkages of human and animal health and environmental changes. We examine underlying elements of the dialogue between science and policy, then we suggest considering SDGs as tool for the service of the environment, wellbeing and justice. We advocate for the translation of planetary health principles into action, together with the consideration of planetary boundaries, to redefine an adaptive environmental law for the sake of social justice and the health of the planet.

**Keywords:** adaptive environmental law; biodiversity; SDGs; health; planetary boundary; planetary health; justice; environment; interdisciplinarity; intersectoriality; boundary-spanning

## 1. Introduction

The 2030 Agenda for Sustainable Development (2015) pledges that no one will be left behind. In a 2018 report, the United Nations Committee for Development Policy warned that "current trends do not point to a degree or speed of advance compatible with the time frame of the 2030 Agenda in some of the fundamental elements that are key to leaving no one behind", including the trends of poverty among others [1].

Yet, immediately prior to the Rio Earth Summit (the United Nations Conference on Environment and Development) in 1992, the World Health Assembly called upon member states to strengthen environmental measures to protect and promote human health, including intersectoral, interdisciplinary approaches emphasizing the protection and promotion of human health and wellbeing, and building on community participation. It also urged participation in international measures for sustainable development that integrate health considerations [2]. The Rio Conference adopted the Convention on Biological Diversity (CBD) together with an action agenda, Agenda 21, which are very comprehensive and encompass socioeconomic dimensions, conservation and environmentally sound management of resources for development, and with an integrated approach at all levels of decision-making (international to local level). While acknowledging the negative effect of the economic policies of the 1980s (the Washington Consensus promoted by the International Monetary Fund, the World Bank, and the United States Department of the Treasury) regarding developing countries, the preamble of Agenda 21 states that international economy should provide a supportive international climate for achieving

environmental and developmental goals. As a basis of action, the agenda noted that the scientific and technological community and policymakers should increase their interaction in order to implement strategies for sustainable development on the basis of the best available knowledge [3]. Indeed, this plan of action gave room for hope.

The foundational principles stated during the Rio Conference, and notably the necessity of a dialogue science/policy for the sake of health and environment, have been reaffirmed several times since then in various international forums.

This study is a qualitative analysis resulting from the content of agendas and strategies regarding biodiversity, health, or sustainable development that have been adopted since the Rio Conference and the adoption of the CBD in 1992. It also builds on the work we conducted on the FutureHealthSEA project and our findings. The point is to determine how these issues have been developed with respect to the dialogue between science and policy, and how the concept of boundary-spanning could help integrate the various aspects (interdisciplinarity, intersectoriality, and communication) needed to improve this dialogue.

While the UN announced that 2020 was a "super year" for nature and biodiversity, the coronavirus pandemic shed a new light on the interlinkages of human and animal health and environmental changes. This study gives an overview of these interlinkages, and aims to assess the effectiveness of the dialogue between science and policy in relation to health and biodiversity in the context of the 2019–2020 coronavirus pandemic.

## 2. Appraisal of the Balance in the Dialogue between Science and Policy

Synergies between international and regional organizations have been promoted for the sake of health and the environment, as shown in the reports of the WHO and the CBD [4], or more recently the Memorandum of Understanding between the FAO, OIE and WHO to organize a joint cooperation and strong focus on antimicrobial resistance in the context of the "One Health" approach [5].

Nevertheless, in retrospect there was no disruption. New agendas and strategies have been adopted since the Rio Conference in 1992, but they contain the same recipes made of general commitments and even the action agenda, Agenda 21, and these kinds of tools lack the explicitness of the "how" to act or to implement these principles. Thus, international commitments remain generic, and most of the time they do not detail means of action to be developed by the government, which leads to issues of translation and implementation at the local level.

### 2.1. A Necessary Boundary-Spanning to Improve the Dialogue between Science and Policy

In the movement toward a disruptive research that could contribute to the redefinition and the transformation of society and political, economic and social systems as they exist, it is crucial to develop boundary-spanning in relation to health, environment and the economic model. Boundary-spanning in the area of social work has been studied by Kerson following an ecological perspective that allows workers to be "less dependent on particular settings and more knowledgeable, adaptable, and flexible" and presupposes thinking systematically [6]. The concept of boundary-spanning has been presented as the tool for "exploration in order to access and obtain knowledge outside local processes" [7]. It thus fosters interconnections of all sorts, whether organizational, institutional or between disciplines, to tackle complex issues. It also gives occasion to think over a new way to do research, notably research on biodiversity, and to conduct it beyond the territories and the sovereignty over those territories by considering the commons. Following the global view of the Sustainable Development Goals (SDGs), we wish to engage in research integrating the notion of planetary boundaries and planetary health, and to open it at the interface of disciplines, as well as political sectors.

### 2.2. More Interdisciplinarity

The coronavirus crisis illustrates the need for integrated research to deal with the causes of the pandemic not only to understand them, but also to delineate which new

trajectory we should take to avoid the reproduction of such a crisis. It necessitates finding another research scheme and to depart from the traditional way of organizing and funding public research. One of the issues in research is linked to the mechanism of research funding. A long-term work based on strong and lasting partnerships is an essential element of research. The length of a multidimensional and incremental research will depend on the obtention of new research projects. Furthermore, if interdisciplinarity is promoted as means to tackle complex issues and to trigger innovation and originality, it is not reflected in the evaluation process, accustomed to assessing scientific work on disciplinary lines [8]. As stated in 2019 by Jean-Pierre Bourguignon, the president of the European Research Council (ERC), "Supporting interdisciplinarity is a challenging obligation because of its remarkable added value" [9]. Nevertheless, the funding of these interdisciplinary projects still has less success compared to those with a narrow disciplinary focus [10].

The interdisciplinary activities of researchers may be difficult to pursue, as they can somehow impede their recruitment for academic positions [11], their publication or even the evaluation of their work, as the recruitment and assessment process of researchers is usually along disciplinary divides. Indeed, it has been underlined that the selection, evaluation and the funding systems in many countries are based on strictly separated disciplines. As an example, we can cite the choice to be made between the 25 panels, and subsections of three main domains, to submit a project for a European Research Council Grant. The same exists for the recruitments of researchers in France, with the existence of specialized scientific commissions within research institutes [12]. As a result, there is a tendency to promote disciplinary researchers over interdisciplinary researchers, their work being evaluated according to discipline-based standards [13]. Examples of these hurdles are even found in environmental sciences, in which the study of multidimensional environmental issues calls for interdisciplinary approaches [14].

### 2.3. A Better Intersectoriality

These issues of research funding and organization are directly related to political choices in terms of flexibility between political sectors, whether at the international level or at the national level. The disciplinary divide existing in research funding also results in a lack of intersectoral dialogue among international or intergovernmental organizations with different mandates, or among government departments or ministries at the national level.

While numerous debates take place between various international actors from different arenas such as WHO, CBD, FAO, OIE, IPCC or IPBES, the lack of genuine intersectoriality often leads to disconnected approaches to complex issues. The intrinsically integrated One Health approach formally recognizes the interconnection between people, animals, plants and their shared environment, and it led to the establishment of a common strategy between the WHO, FAO and OIE to coordinate global activities to address health risks at the animal–human–ecosystem interfaces [15]. The same actors organized the Global Early Warning System for Major Animal Diseases, Including Zoonoses [16], which "embodies a cross-sectoral and multidisciplinary collaborative tool in addressing health risks at the human-animal-ecosystem interface", transformed in 2011 into GLEWS+ [17] with the aim to:

"1. Systematically link to areas such as wildlife health, food and biological threats;

2. Drive more advanced and cross-sectoral risk assessment when a need is identified; and

3. Provide more opportunities for participation by a broader range of stakeholders via specific working groups established on priority areas".

Nevertheless, this global early warning system does not seem to be very active if we consider the last updates on the GLEWS+ website and the absence of references to joint publications, whether on this website or on the website of the three organizations that are part of that system. In particular, in relation to the coronavirus crisis, we could not find any sign of activity of this system, while a One Health approach, joining the forces of various

international organizations around WHO, could make a difference in terms of international coordination [18].

Many reasons could explain this lack of coordination, and they are significant when it comes to understanding the limit of interventions of international organizations or intergovernmental platforms such as IPCC or IPBES. Among them, we can signal the lack of means for an effective coordination. As in the case of research projects, the long-term maintenance of such an initiative is crucial to its success. Sometimes joint actions initiated by international or regional organizations are created ad hoc to celebrate an anniversary (like Earth Day), but they are not lasting as they were planned to because of staff turnover, lack of funding or new priorities taking place of the older ones.

This insufficiency of resources also affects governments within international or intergovernmental organizations, and while these huge structures should allow a better coordination among the members, they may be unable to change their trajectories, which should be aligned with SDGs and other international commitments, or to react fast in case of an emergency.

In the case of the coronavirus pandemic, countries have been deciding in isolation the measures to be taken in the course of the progression of the epidemic nationally, without procedures guidelines and help from the WHO. This absence of coordination between countries, which could have been orchestrated by the WHO, is underlined regarding the European Union in the joint motion for a resolution on EU-coordinated action to combat the COVID-19 pandemic. The motion highlights that EU response has been "marked by a lack of coordination between Member States in terms of public health measures, including restrictions on the movement of people within and across borders and the suspension of other rights and laws". It notably calls for "a coordinated post-lockdown approach in the EU, in order to avoid a resurgence of the virus" [19]. It insists on the obligation of solidarity among Member States, as well as solidarity of the entire international community and a strengthening of the UN system and WHO.

*2.4. Communication Issue: Scientific Debate and How to Share Complex Results*

The research questions at stake are not easily understandable or accessible. In the media, researchers are called in as experts, often from the same small pool of researchers, to answer a very specific question in their own discipline. The cover of a multidimensional topic in this case results in the juxtaposition of narrow and specific expertise when it would be necessary to broaden the perspective, step back and offer a critical thought regarding the whole situation. It should present scientific controversy and uncertainty through the lens of science and keep it in its whole context, not as mere opinions or approximations to cause a juicy polemic and thus maximize the audience.

In this context, media could take advantage of the qualities of researchers such as their openness, emotional stability, curiosity and eccentricity [20] to communicate, with their help, scientific knowledge in favor of a better dialogue between science and policy or science and society.

One of the issues resides in the fact that researchers may have difficulties communicating complex and interlinked results and presenting uncertainties in an intelligible way for the public [21]. The mode of communication is an issue, and it should be adapted to the specific audience it targets, whether it is results to inform the society or policymakers, or for a specific question in relation to a decision to make.

## 3. SDGs: From a Final Goal to a Tool for the Service of the Environment, Wellbeing and Justice

As stated previously, boundary-spanning is needed across disciplines, territories and organizations to tackle complex issues such as environmental changes. However, what the coronavirus crisis has shown us is that it is imperative to depart from the daily humming of the functioning of our societies.

After the Rio Conference in 1992, if the expression "sustainable development" became very popular in the international arena, in response to various agendas, as noted by Adams,

"Rio offered no decisive breakthrough in business as usual, for governments, business or citizen" [22] In preparation for the Johannesburg Conference on Sustainable Development in 2002, the United Nations Secretary General K. Annan declared that "so far our scientific understanding continues to run ahead of our social and political response". He added, "With some honorable exceptions, our efforts to change course are too few and too little. The question now is if they are also too late" [23].

In relation to the coronavirus crisis, and to environmental and climate change, this question resonates ominously. The 2030 Agenda for Sustainable Development was presented has a plan of action for people, planet and prosperity, but so far, we have not been able, at the planetary level, to curb the massive damages caused to the environment, which may result in a sixth mass extinction of animal and plant species [24,25], or to protect the ecosystem and human health, as shown by the coronavirus pandemic.

To depart from the "business as usual" solution, we consider using SDGs that have been debated, integrated into national contexts and assessed at the national and regional level or within private companies, not as final goals, but as a tool to be combined in order to reach more disruptive final goals based on the state of the environment and putting planetary health at the center.

We identified issues with the actual SDGs and Aichi biodiversity targets (directly linked with SDGs) that might impede their accomplishments. Previous to the adoption of a new strategy for biodiversity and new biodiversity targets, we think that, based on these issues and acknowledging that the coronavirus crisis worldwide, this is the result of unprecedented environmental crisis.

There is a lack of clear objectives regarding the links between health and biodiversity in SDGs and the Aichi targets defined in accordance with SDGs.

SDGs have been drafted to be more comprehensive than the Millennium Development Goals, with more targets selected for their clarity and measurability and the fact that they are "aspirational yet attainable" and nationally relevant and adaptable, as well as evidence- or science-based and adjustable [26]. Biodiversity is supposed to be a crosscutting issue into SDGs, and a technical note of the CBD states that point is intended to help decision-makers to understand more easily the contributions of biodiversity in achieving SDGs in mapping of the linkages between SDGs, the Strategic Plan for Biodiversity 2011–2020 and its 20 Aichi biodiversity targets [27]. SDG 3, "Ensure healthy lives and promote wellbeing for all at all ages", is significant in this regard. Indicators associated with this goal are mainly health indicators developed by the WHO. Indeed, in its technical note the CBD only refers to indicator 3.9 in relation to different types of pollution having an effect on health. The note also stipulates that indirect links exist between biodiversity and human health, giving the example of the role of diverse agricultural ecosystems in contributing to sustainable production increases and reducing the use of pesticides and other chemical inputs, all of which can have positive impacts on human health. Nevertheless, this point is not embedded in SDG 3, and thus biodiversity and ecosystem functioning are kept at the margin of health issues and not acknowledged as an integral part of the reflection on the improvement of health worldwide.

This results in a lack of clear objectives for health and biodiversity combined, and the technical note and policy brief on biodiversity and the 2030 Agenda for Sustainable Development is somehow an attempt to address this intrinsic deficiency of SDGs. This endeavor cannot entirely be successful as the role of biodiversity as not been put at the core of the drafting of SDGs.

While the new Post-2020 Global Biodiversity Framework is being discussed in order to be adopted during the next Conference of the Parties to the CBD, the difficulties or even impossibility to achieve the Aichi targets and SDGs might discredit the implementation of new biodiversity objectives (for a midterm assessment of Aichi, see [28]). It might also undermine the willingness of many countries or local communities within countries to commit to vague targets [29].

The issue of confusion between indicators, targets and objectives has already been raised regarding SDGs and Aichi targets, as well as the lack of numerical outcome, specific deadline and defined domain for many of the targets. While the new post-2020 biodiversity targets are expected to be science- and knowledge-based and "SMART" (specific, measurable, ambitious, realistic and time-bound) [30], we have seen with the Aichi targets that the absence of clarity between objectives, targets and indicators led to a difficult implementation at the national level [31].

Thus, some SDGs could be used as tools to reach other, more integrative SDGs such as SDG 10 (reducing inequalities), which could be realized by monitoring multiple SDGs such as SDG 3 (health), SDG4 (education) and SDG5 (gender equality). As shown by Vandemoortele [32], inequality is not addressed by SDGs, which instead mention extreme poverty to reduce inequalities and establish from the start performance targets for developing countries and vaguely defined targets for developed countries. Thus, the agenda offered by SDGs should be universal, and the countries where SDGs are implemented at the national level should be supported by international organizations to set up national priorities adapted from SDGs for their national situation. This would then be a way to integrate the realization of SDGs in the national objectives of biodiversity coming from the post-2020 biodiversity strategy, and to consider altogether health and biodiversity issues in national law and public policies.

### 3.1. From Commitment to Concrete Involvement

The genuine consideration for ecosystem health, an analogy with human health that applies to the health of the environment, allows assessing the health of the environment and reaching environmental objectives using "health" criteria [33]. It gives the opportunity to consider the socio-ecosystems and the interactions between sociocultural practices and animal and human health, as well as environmental health [34]. In order to succeed in a concrete consideration of these links, the dialogue between science and policymakers is crucial. It implies the transformation of the political commitment to a concrete involvement: going from strategy to strategy, knowing at the time of their implementation at the national level that they will fail because of the delay between the international commitment and its national translation is not an option anymore. We should step back, change the rules and start to act simultaneously both internationally and locally in a genuine way, and not just in order to tick off lists to report a good formal advancement toward the objectives.

### 3.2. Adaptive Laws and Policies

In order to respond in an appropriate manner to the environmental emergency, we need adaptive laws that create a framework able to integrate new objective knowledge of the environment and health in order to produce evidence-based measures or policies adapted to the situation in specific socio-ecological systems, or socioecosystems [35]. The group working on SDGs called for nationally adaptable and adjustable targets as science advances, or when countries choose to raise the level of ambition. This implies that national laws should have the same properties as those SDGs, and should contain in their drafting the possibility of a cursor that moves with advances in science (ecology, medicine, agronomy, etc.). Environmental law needs to be rebuilt around the notion of science-based knowledge relying on planetary limits. In this respect, we call for putting planetary health principles into action.

### 4. Putting Planetary Health Principles into Action

The report of the Rockefeller Foundation–Lancet Commission on Planetary Health defined planetary health as "the achievement of the highest attainable standard of health, wellbeing, and equity worldwide through judicious attention to the human systems—political, economic, and social—that shape the future of humanity and the Earth's natural systems that define the safe environmental limits within which humanity can flourish" [36]. It referred to the planetary boundaries proposed by the Stockholm Resilience Center [37,38],

quantitative boundaries associated with biological and physical processes and systems allowing the maintenance of the Earth's functions.

For a practical action that takes the limits of the planet into account, we suggest the integration of health issues within planetary boundaries that relate to human-induced changes to the environment. For a real leverage of the actions taken in relation to health, as in the case of a pandemic, for instance, we need strong collaborations beyond national boundaries to activate coordinated efforts to protect people worldwide from global threats through the general improvement of health and the reduction of disparities [39,40]. As already proposed by Gostin in preparation of negotiation of the SDGs, we need to integrate global health with justice, and put the common efforts for the reduction of health disparities between the well-off and the poor, thanks to the provision of public health services to the whole population, universal public health coverage and the improvement of socio-determinants of health [41] and of socio-environmental determinants of health. These determinants are strongly linked to environmental justice, which is too often belittled by environmental law [42] A way to reconcile the fight for environmental justice and public health would be, legally speaking, to rely on human rights whether at the international, regional or national level, and particularly on the right to health and the right to a healthy environment currently under scrutiny by the United Nations General Assembly. This latter will help to increase the role of the public in environmental governance [43].

### 4.1. Indicators and Objectives

In relation to planetary boundaries, it is crucial to involve corporates in this system for the evaluation of the transboundary effect of their activities. It calls for a real activation of environmental law with a contribution of corporates like the one reflected by the United Nations Global Compact initiative, which supports companies doing business responsibly by aligning their strategies and operation principles on human rights, labor, environment and anticorruption, and taking strategic actions to advance broader societal goals such as SDGs. The objectives should rely on staying within the nine planetary boundaries and the various suggestions made by scientists for key aspects such as the global supply chain, or more generally, the integration of planetary-boundary indicators within the life-cycle assessment of manufactured products [44], indicators for business to guide investments, innovation and performance indicators along the value chain [45] or in relation to biodiversity with healthy ecosystem metrics [46].

### 4.2. Monitoring

Once they are adopted, these quantitative indicators can be monitored and assessed through a form of international and national evaluation of planetary limits, which implies and international involvement and the recognition of a united community at the international level acting in solidarity. There is a need for improved cooperation between various sectors, including health, environment, energy, agricultural and transport, as well as chemical and other industries, as stated in the Helsinki Declaration to protect human and planetary health for the 2020s [47]. Monitoring and assessment can also help refine the definition of planetary boundaries by the addition of planetary sub-boundaries, like in the case of a water planetary boundary to complement existing tools for water-resource management, with an approach for assessing water-cycle modifications as part of the wider human impact on the Earth system [48].

## 5. Conclusions: Justice at the Core in Favor of the Environment and Wellbeing

To start with the planetary limits instead of the development goals invites constant acknowledgement of the limits of action of humanity, taken as a whole. This, together with the assessment of the environmental effects of the lockdown (even during a short period of time) due to the coronavirus pandemic, must lead us to adopt major political and economic measures centered on the protection of the environment and the wellbeing of people.

Keeping in mind the urgency to reduce inequalities and disparities on the planet to protect health and the environment, by the same token, we suggest turning to planetary boundaries and transforming environmental law using these boundaries as cursors for environmental measures and penalties, whether at the global or local scale. The planetary boundaries would serve to define SMART indicators based on cursors of the state of the planet within these boundaries. Boundary-spanning in this case would mean to look after the interactions between these boundaries to avoid the amplification of a phenomenon across boundaries.

For instance, the ecocide movement already refers to planetary boundaries, with ecocide being defined as "the extensive loss or damage or destruction of ecosystem(s) of a given territory, whether by human agency or by other causes, to such an extent that peaceful enjoyment by the inhabitants of that territory has been or will be severely diminished" [49].

The integration of the planetary-boundary dimension into environmental law would strengthen its scientific ground and participate in its adaptivity and necessary dynamic basis. It would be a way to promote actions undertaken simultaneously at various levels of decision-making [50] while considering disparities around the globe in order to reduce them. It would acknowledge a multilevel environmental governance, for the sake of environmental and social justice, and eventually the health of the planet.

**Author Contributions:** S.M. and C.L. conceived the ideas and secured funding; C.L. wrote the manuscript and provided expertise; S.M. provided expertise and feedback. Both authors have read and agreed to the published version of the manuscript.

**Funding:** This work is a contribution to the FutureHealthSEA project (Predictive scenarios of health in Southeast Asia: linking land use and climate changes to infectious diseases) funded by the French ANR (grant number ANR 17 CE35-0003). (http://www.agence-nationale-recherche.fr/Project-ANR-17-CE35-0003) (accessed on 13 April 2021).

**Institutional Review Board Statement:** Not applicable.

**Informed Consent Statement:** Not applicable.

**Data Availability Statement:** Data sharing not applicable.

**Conflicts of Interest:** The authors declare no conflict of interest.

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
