# Peer review of "Biodiversity Targets, SDGs and Health: A New Turn after the Coronavirus Pandemic?"

_sustainability, doi:10.3390/su13084353_

Round 1

Reviewer 1 Report

The presented manuscript is of an overview character. It presents a very interesting and topical subject. I found many accurate observations and comments regarding the described reality. I believe this text can be an interesting foundation for considering the Circular Economy. However, I missed a presented thesis and research question. The author has built a manuscript based on the topics described. However, the author has substantiated the conclusions with an extensive literature review.

I have two specific comments / questions.

  • Line 103-106 - I do not understand this dependency. Maybe it’s worth explaining?
  • Line 176-177 - I don’t think you can agree with such a harsh statement.

Overall, I rate the article as very good.

Author Response

Answers to Reviewer 1

Thank you very much for this useful review which helped us developing our ideas. We modified the manuscript according to your remarks.

About the general comment on the manuscript and the lack of research question and detailed methodology, we have added elements at the end of the introduction:

This study is a qualitative analysis resulting from the content of agendas and strategies regarding biodiversity, health or Sustainable Development that have been adopted since the Rio Conference and the adoption of the CBD in 1992. It also builds on the work we conduct on the FutureHealthSEA project and our findings. The point is to determine how these issues have been developed with respect to the dialogue between science and policy and how the concept of boundary spanning could help integrating the various aspects (interdisciplinarity, intersectoriality and communication) needed to improve this dialogue.

While the UN environment announced that 2020 was a Super Year for nature and biodiversity, the coronavirus pandemic shed a new light on the interlinkages on the human and animal health and environmental changes. This study gives an overview of these interlinkages and aims to assess the effectiveness of the dialogue between science and policy in relation to health and biodiversity in the context of the 2019-2020 coronavirus pandemic.

About the specific comments / questions.

  • Line 103-106 - I do not understand this dependency. Maybe it’s worth explaining ? We have developed this point and added a few references.

Indeed, it has been underlined that the selection, evaluation and the funding systems in many countries are based on strictly separated disciplines[1] (Muller et al. 2019). As a result, there is a tendency to promote disciplinary researchers over interdisciplinary researchers, their work being evaluated according to discipline-based standards (Klein et al. 2016).

  • Line 176-177 - I don’t think you can agree with such a harsh statement.

Yes, thank you for pointing this out. The statement has been modified.

One of the issues resides in the fact that researchers may have difficulties to communicate complex and interlinked results and to present uncertainties in an intelligible way for the public (Olson, 2018).

[1] As an example, we can cite the choice to be made between the 25 panels, sub-section of 3 main domains to submit a project to a European Research Council Grant. The same exists for the recruitments of researchers in France with the existence of Specialized Scientific Commissions within research institutes.

Reviewer 2 Report

The article is of great relevance in times where research and academia are asked to critically engage with the topics of the Agenda 2030. The research is of good quality and produces knowledge that is applicable outside the field of expertise. The reflections shared have many contact points with the legal science, for example, where there is an increasing need to adopt integrated solutions to complex problems. Furthermore, the results have potential application for future policies rooted around the theme of planetary health and project applications connected to the thematic area.

Author Response

Thank you very much for your very encouraging comments, we are very grateful.